# Neurodevelopmental Outcome of Very Low Birth Weight Infants in the Northern District of Israel: A Cross-Sectional Study

**DOI:** 10.3390/children10081320

**Published:** 2023-07-31

**Authors:** Michal Molad, Ayala Gover, Zaki Marai, Karen Lavie-Nevo, Irina Kessel, Lilach Shemer-Meiri, Marina Soloveichik

**Affiliations:** 1Neonatal Intensive Care Unit, Carmel Medical Center, Haifa 3436212, Israel; michize@gmail.com (M.M.); klavie@gmail.com (K.L.-N.); irinakessel@gmail.com (I.K.); marinaso@clalit.org.il (M.S.); 2Rappaport Faculty of Medicine, Technion–Israel Institute of Technology, Haifa 3525433, Israel; lilachshemer@gmail.com; 3Neonatal Intensive Care Unit, Bnai Zion Medical Center, Haifa 31048, Israel; 4Pediatrics Department, Carmel Medical Center, Haifa, 3436212, Israel; marai_zaki@yahoo.com

**Keywords:** neurodevelopmental outcome, VLBW, preterm, cerebral palsy

## Abstract

Background: Currently, no local database in Israel collects neurodevelopmental outcomes of very low birth weight (VLBW) preterm infants. We investigated neurodevelopmental outcomes in one district of the largest healthcare organization in Israel. Methods: A cross-sectional study including all VLBW (<1500 g) preterm infants born between 1 January 2006 and 31 December 2016 who were followed in any of seven child development centers in Israel’s Northern District. Data were retrospectively collected from the computerized medical record database. Results: Out of 436 participants, 55.1% had normal developmental outcomes. A total of 8.9% had cerebral palsy (CP), 12.2% had a global developmental delay (GDD), and 33.4% had a language delay. Out of the extremely preterm infants (*n* = 109), 20.2% had CP, 22.0% had GDD, and 44.9% had language delay. We found a statistically significant higher rate of abnormal neurodevelopment outcomes in non-Jews compared to Jews (57% vs. 37.8%, respectively, *p* < 0.0001). Conclusions: We found a relatively high overall rate of CP in our local population and a significant difference in neurodevelopmental outcomes between Jews and non-Jews. This study emphasizes the need for an expanded and detailed national database collecting post-discharge outcomes, as well as an assessment of national healthcare resource allocation and inequalities in preterm infants’ post-discharge care.

## 1. Introduction

Preterm infants are at high risk for short and long-term neurodevelopmental impairment in the neuro-motor, cognitive, language, and behavioral domains [1]. The estimated prevalence of cerebral palsy (CP) in preterm infants is around 15% in infants born at 22–27 weeks of gestation, and it decreases with increasing gestational age (GA) [2,3]. Language and speech delays occur in around 30% of very preterm infants examined at preschool age [3]. Other neurodevelopmental sequelae in children born preterm include cognitive delay, milder motor impairments (e.g., developmental coordination disorder (DCD)), hearing impairment, autism, and attention deficit and hyperactivity disorder (ADHD) [4,5]. 

Over the past decades, rates of survival of extremely preterm infants have increased with no consistent improvement in neurodevelopmental outcomes. The greatest increase in survival rates occurred in extreme preterm infants born before 25–26 weeks of gestation [6], probably due to changes in practice, as well as increasing rates of active treatment. However, increased survival of these extremely preterm infants may go hand in hand with a rise in morbidities and impairments, which are closely related to lower GA at birth. 

While in some countries, rates of CP are decreasing [7], in others, they remain constant [8,9]. As rates of survival with and without morbidities vary significantly between countries, it is necessary to evaluate local rates of survival and neurodevelopmental outcomes to facilitate perinatal decision making and parent counseling in cases of extreme preterm deliveries. In Israel, a small country with a high fertility rate [10], around 1% of all liveborn deliveries are of infants with a birth weight below 1500 gr. This population has an overall mortality rate of 14.2% in the first year of life and a much higher rate in the group of extremely preterm infants born before 28 weeks. Nevertheless, our local rates of survival of these extreme preterms have improved over the years, and from 2000 to 2017, there has been a persistent decline in mortality from 49% to 36.7% in the first year of life [11]. Increased survival may be related to better obstetric antenatal care and improved ability in resuscitation and therapies for the newborn in modern neonatology, and may raise the issue of extending the limit of viability to 22–23 weeks of gestation, in which routine resuscitation is not currently performed in Israel. However, as one of the ethical principles that apply to all medical care is nonmaleficence, avoiding harm, the potential for survival with severe disabilities and a high morbidity burden on the child and family should be carefully studied and considered in treatment decision making as well as for the anticipation and meeting of this population’s needs by the public health system.

The Israeli national database collecting data for very low birth weight (VLBW) preterm infants currently does not include post-discharge neurodevelopmental outcome data, and, to our knowledge, no local study exploring outcomes has been performed as yet. 

In this cross-sectional study, we investigated the neurodevelopmental outcome of children aged 3–13 years who were born preterm with VLBW (<1500 g) and were followed in one district of the largest healthcare organization in Israel, and examined the differences in outcomes between gestational age, weight, and ethnicity groups. 

## 2. Materials and Methods

Participants: “Clalit” Health Services (CHS) is the largest of four integrated healthcare organizations in Israel. The study population included all VLBW (<1500 g) preterm infants born between 1 January 2006 and 31 December 2016 who were followed in any of the 7 CHS child development centers in Israel’s Northern District—Haifa and Western Galilee county. Haifa is the 3rd largest city in Israel and has a large metropolitan area. In Western Galilee, most of the population lives in small villages, but there are a few large cities. Most of the population in Haifa is Jewish, and around 20% are Arab, both Christian and Muslim, as well as Druze [12]. In Western Galilee, there is a non-Jewish majority (>50%) consisting mostly of Arab Muslims, with some Druze and Arab Christians. Between 50 and 70% of the population in these areas is insured by CHS [13]. 

At the time of data collection, participants were between 3 and 13 years of age. Infants who were lost to follow-up and infants with major congenital abnormalities (i.e., structural anomalies that have significant consequences on the infant or require intervention) or chromosomal anomalies were excluded. The study was approved by CHS’s Ethical Board.

Data collection: Data were retrospectively collected from the computerized medical record database of CHS child development centers in Haifa and Western Galilee County. Demographic and perinatal data were extracted, including GA at birth, place of birth, mode of delivery, birth weight, ethnicity, etc., as well as complications of prematurity, including necrotizing enterocolitis (NEC), late-onset sepsis (LOS) patent ductus arteriosus (PDA), intraventricular hemorrhage (IVH), white matter injury (WMI), posthemorrhagic hydrocephalous (PHH), and retinopathy of prematurity (ROP), defined by commonly used published criteria [14,15,16,17]. Bronchopulmonary dysplasia (BPD) was defined as preterm infants born <32 weeks of gestation requiring supplemental oxygen for at least 28 days and at 36 weeks postmenstrual age [15], without stratifying by severity. Low-grade IVH was defined as grade I/II and high-grade IVH was defined as grade III/IV by Papile grading [14]. WMI was defined as low grade when periventricular echogenicities were observed for >7 days, and high grade (cystic WMI) was considered when there were cystic lesions (grade II-IV by DeVries classification [17]). Small for gestational age (SGA) was defined as birth weight below the 10th percentile for gestational age by Fenton growth charts for boys and girls, appropriate for gestational age (AGA) was defined as birth weight between 10th and 90th percentile, and large for gestational age (LGA) was considered above the 90th percentile. Center of birth size was defined by the annual number of deliveries, so centers with over 4000 deliveries per year were considered large, and those with less than 4000 deliveries a year were considered small/medium. Neurodevelopmental outcome measures included the presence of CP, language delay, global developmental delay (GDD), autism spectrum disorder (ASD), developmental coordination disorder (DCD), and attention deficit and hyperactivity disorder (ADHD). The following definitions and tests were used: language delay was assessed by using the Katzenberger Hebrew Language Assessment for Preschool Children (KHLA) [18] or the Preschool Language Scale Fourth Edition (PLS4 HEB) [19]. GDD was assessed by using the developmental Quotient assessment. ASD was assessed by Adaptive Behavior Assessment System II Parental Questionnaire (ABAS II) [20] and by using the Diagnostic and Statistical Manual of Mental Disorders (DSM 5) in combination with neurodevelopmental examination, dyadic game observation, and kindergarten staff report. DCD was assessed by using the Developmental Coordination Disorder Questionnaire (DCDQ) [21] and Movement Assessment Battery for Children (M-ABC) [22]. CP was diagnosed by evaluating clusters of specific clinical CP signs of a static, not progressive motor deficit without evidence of a neurodegenerative disorder corresponding to CNS imaging findings, following the guidelines of the American Academy of Neurology and Child Neurology Society [23]. It was categorized as diplegic, hemiplegic, or quadriplegic. Hearing and visual outcomes were collected as well. The most recent available neurodevelopmental diagnosis in the chart was used. 

Data analysis: Continuous variables are presented as means and standard deviations or as medians and ranges. Categorical variables are presented as percentages. The association between developmental outcomes and demographic and clinical characteristics was analyzed using the chi-square test for the categorical variables and the independent t-test or Mann–Whitney, as appropriate, for the continuous variables. Multivariable logistic regression was used to identify the factors associated with complications of prematurity. Odds ratios with 95%CI are presented. *p* < 0.05 was considered statistically significant. All analyses were performed using IBM Statistics version 24 (SPSS). 

## 3. Results

There were 469 VLBW preterm infants registered to any of the seven child development centers of the county. Twenty-eight were lost to follow-up, and four had major congenital abnormalities; one was due to congenital CMV infection, one had VACTERL association, one had polymicrogyria, and one had high suspicion of an undiagnosed congenital syndrome. Another child died after 2 years of follow-up (unknown etiology). A total of 436 infants comprised the final study cohort. In four infants, there were some incomplete data on the presence or absence of complications of prematurity. Demographic characteristics are presented in Table 1. Gestational age at birth ranged between 23 and 37 weeks of gestation, and birth weight ranged from 450 to 1495 g. All infants were born in hospitals with third-level NICUs, and there were eight different birth centers. At the time of data collection, 257 (58.9%) of the participants were between 3 and 6 years of age, 131 (30%) were between 6 and 9 years of age, and 48 (11%) were between 9 and 13 years. Timing, number, and frequency of follow-up visits varied between patients, as there are no current guidelines in Israel defining specific time points for neurodevelopmental follow-up after 18 months of age, and they are rather performed in intervals determined individually according to the child’s condition and needs, as well as the child development center’s availability. 

Out of all 436 patients, 55.1% had normal developmental outcomes. A third of the children had a language delay, 12.2% had a global developmental delay, 8.9% were diagnosed with CP, 7.8% had ADHD, and 1.6% were diagnosed with ASD (Table 2).

There was no association between neurodevelopmental outcomes and the center of birth or mode of delivery. There were three large birth centers with over 4000 annual deliveries and five small/medium centers with less than 4000 annual births. No significant differences were found in neurodevelopmental outcomes between infants born at large birth centers versus infants born at small/medium birth centers. 

There was considerable year-to-year variation in rates of all neurodevelopmental outcomes over the studied 10-year period with no consistent trend of rise or decline in any of the outcomes (Appendix A).

As expected, abnormal neurodevelopmental outcome was associated with low birth weight and GA at birth, and this was statistically significant (*p* = 0.0001 for both). Only 40.3% of the preterm infants born between 23 and 27 + 6 weeks of gestation had normal development, compared to 57.7–59.1% of normal development in the other GA groups. Rates of neurodevelopmental impairments are presented by gestational age categories (extremely preterm infants < 28 weeks GA, very preterm infants 28–31 + 6 weeks of GA, and moderate–late preterm infants 32–37 weeks of GA) (Table 3).

CP was significantly associated with younger GA (*p* < 0.0001) as well as with lower birth weight (*p* = 0.018). Out of all children with CP (N = 39), 56.4% were born at 23–27 + 6 weeks of gestation, 33.3% were born at 28–31 + 6 weeks, and 10.2% were born after 32 weeks. The rate of CP was 20.2% at 23–27 + 6 weeks of gestation (27.4% in infants < 26 weeks), 5.7% at 28–31 + 6 weeks, and 4.1% after 32 weeks of gestation. There were no cases of CP after 34 weeks. CP was associated with NEC and BPD (*p* = 0.012, *p* < 0.0001, respectively) after controlling for birth weight and gestational age, as well as with IVH, WMI, and PHH (*p* < 0.0001 for all). 

Language delay and GDD were significantly associated with younger GA (*p* = 0.007 and *p* = 0.001, respectively); however, autism spectrum disorder and hearing impairment or deafness were not. Language delay was also associated with BPD (*p* = 0.003), but this effect was lost after controlling for birth weight. 

Sonographic findings of high-grade IVH, cystic WMI, and PPH were significantly associated with CP, GDD, and language delay but not with ADHD or autism (Table 4).

Considering the weight percentile for gestational age, the rate of CP was lower in SGA infants (2/89, 2.2%) compared to AGA (35/341, 10.3%) and LGA (2/6, 33%), and this was statistically significant (*p* = 0.014); however, when adjusting for GA, this effect was lost. Language delay occurred at a higher rate in LGA versus SGA and AGA infants (83.3%, 38.2%, 31.3%, respectively, *p* = 0.014), but this finding did not remain significant after adjusting for GA as well. The rates of DCD, GDD, ADHD, and autism were not related to the weight percentile group.

Sex differences were found in some of the neurodevelopmental outcomes. Males were more likely to be diagnosed with CP than females (11.8% in males and 6.0% in females, OR 2.09 (95% CI 1.04–4.19), *p* = 0.034,), as well as ADHD (11.4% in males and 4.2% in females, OR 2.94 (95% CI 1.34–6.47), *p* = 0.005), and language delay (40.8% in males and 25.9% in females, OR 1.97 (95% CI 1.31–2.96), *p* = 0.001). There were no sex differences in rates of GDD, DCD, and autism.

Comparing outcomes between Jews and non-Jews (mostly of Arab or Druze descent), we found strong evidence to suggest differences in abnormal neurodevelopmental outcomes, with non-Jews more likely to be abnormal compared to Jews (57% in non-Jews vs. 37.8% in Jews, OR 2.17 (95% CI 1.4–3.2), *p* < 0.0001), as well as in the rates of CP (12.3% in non-Jews vs. 6% in Jews, OR 2.17 (95% CI 1.1–4.3), *p* = 0.022,), GDD (17.7% in non-Jews vs. 7.7% in Jews, OR 2.5 (95%CI 1.4–4.69), *p* = 0.002), and language delay (41.7% in non-Jews vs. 26.8% in Jews, OR 1.95, (95%CI 1.3–2.9), *p* = 0.001). Only DCD was less likely to be diagnosed in non-Jews than in Jews (1.1% in non-Jews vs. 5.7% in Jews, OR 0.18 (95% CI 0.04–0.8) *p* = 0.012). The mean number of visits to the child developmental center was slightly lower in non-Jews compared to Jews; however, this was not statistically significant (25.5 ± 49 vs. 27.1 ± 35, respectively, *p* = 0.059).

In a multiple logistic regression model adjusted for all variables with *p* < 0.1 in the univariate analyses (GA and birth weight, weight percentile for gestational age, NEC, BPD, PDA, LOS, IVH, and WMI, ethnicity, and sex), there was a significant association between any abnormal neurodevelopmental outcome and male sex (*p* < 0.001, OR 2.1 (95% CI 1.4–3.2)), non-Jewish ethnicity (*p* < 0.001, OR 2.2 (95%CI 1.5–3.4)), cystic WMI (*p* < 0.001, OR 15.4 (95%CI 1.9–123.8)), and high-grade IVH (*p* = 0.007, OR 8.1 (95%CI 1.8–37.7)), but not with other complications of prematurity. The same model found an association between GDD and cystic WMI (*p* < 0.001, OR 26.5 (95%CI 7.0–99.9)), high-grade IVH (*p* < 0.001 OR 9.7 (95%CI 2.8–33.1)), large for gestational age (*p* = 0.023, OR 9.4 (95%CI 1.4–65.5)), and NEC (*p* < 0.001, OR 6.01 (95%CI 2.3–15.7)). 

## 4. Discussion

Preterm infants are a vulnerable population at high risk for mortality, morbidity, and neurodevelopmental impairments that carry lifelong consequences. 

In this study, we present the local rates of neurodevelopmental impairments in a large district in Israel. We found a relatively high rate of CP in our preterm population, even when considering the variations in the prevalence of CP across studies. Our overall local rate of 8.9% is higher than reported by other studies utilizing global data. Oskoui et al. found a prevalence of 6% in infants born less than 1500 g in their meta-analysis [24]. In the 23–26 weeks gestation group, our rate was 27.4%, close to the rate reported by Oskoui et al. in this group (26%); however, in our study, there were only two infants born at 23 weeks, which should have lowered the rate. Another meta-analysis by Himpens et al. [2] report a much lower rate (16.6%) in infants born at 23–27 weeks gestation, and Jarjour et al. [25] report a rate of 9–18% in infants born at less than 25 weeks gestation. More recently, Bell et al. [6] reported a rate of 8.4% of moderate to severe CP in infants born at 23–26 weeks of gestation between the years 2013 and 2016 and 9.5% of mild CP.

The reason for our higher rate of CP is unclear. While CP in preterm infants is multifactorial, associations with chorioamnionitis, birth asphyxia, BPD, PDA, and severe brain lesions have been previously described [26,27,28,29]. In our cohort, there was a significant association between CP and BPD, NEC, and brain injury, but the rates of these complications of prematurity were comparable with those described in the literature. However, we could not account for other influencing factors, including chorioamnionitis, birth asphyxia, postnatal steroids and BPD severity, neonatal surgery, antenatal magnesium sulfate neuroprotection, and caffeine treatment [29]. 

Language delay is common in preterm infants. Unlike the rate of 5–12% in children aged 2–5 years in the general population [30], Stipdonk et al. described a rate of 22–45% of language problems in children born before 32 weeks of gestation [31]. Foster-Cohen et al. reported a prevalence of 30% language delay in children born before 33 weeks of gestation. [32]. Our local rate of language delay was comparable to previously published data (33.4%), emphasizing the need for a careful and thorough speech and language evaluation for every child born very preterm. 

Unlike some previous studies suggesting a negative impact of SGA on neurodevelopment [33,34,35], there was no significant effect of weight percentile for gestational age on any of the neurodevelopmental outcomes in our cohort, even though 20% of our population were SGA infants. A large systematic review and meta-analysis [35] including over 52000 children found poorer cognitive outcomes in children born SGA in both the term and preterm groups compared to children who were born AGA; however, the effect size was smaller for the preterm-born group compared to the term-born group. In another population-based matched case–control study [34], the risk of CP was inversely related to birth weight even outside the range considered SGA in both term and preterm infants. Other studies did not support this finding in preterm infants, although it was observed in the term population [36]. The proposed mechanisms for this link are a suboptimal in utero environment, placental insufficiency, and in utero undernutrition affecting brain development through epigenetic changes and fetal programming, as well as rendering the preterm SGA infant more vulnerable to subsequent postnatal insults [35,37]. In our cohort, we could not differentiate between infants who had intrauterine growth restriction with evidence of abnormal placental function and “constitutionally” small infants, but lower scores in multiple cognitive domains were previously described in SGA term infants versus AGA infants, even when the placental function was normal, as reflected by umbilical artery Doppler [38]. Possibly, the effect of SGA in our cohort was counterbalanced by the relatively advanced gestational age of some of the SGA group, as all infants in our cohort who were born at 33–37 weeks were essentially SGA due to our inclusion criteria of birth weight of <1500 g. Interestingly, LGA and not SGA were associated with global developmental delay in multiple logistic regression. Several previous studies raised concern regarding poorer neurodevelopmental outcomes in both ends of abnormal intrauterine growth [39,40] and hypothesized that the increased risk of neurodevelopmental impairment in LGA infants may be affected by maternal conditions such as gestational diabetes. Other studies did not show adverse neurodevelopmental outcomes in LGA preterm infants [41]. Since our cohort included a very small sample size in this group (n = 6), this association may represent a chance finding. 

Severe brain lesions were in correlation with CP, GDD, and language delay, consistent with previous reports [42,43]. In one report, 92–96% of children born preterm prior to 36 weeks GA who were later diagnosed with CP had brain abnormalities detected by ultrasound during the neonatal period, and 83–89% of the ultrasound findings were major abnormalities, including severe IVH, cystic WMI, basal ganglia lesions, and focal infarction. Cystic WMI was the most predictive of CP [44]. Magnetic resonance imaging studies further confirm the link between white matter injury with cognitive and language impairments [45,46]. 

We found sex differences in the rate of CP, ADHD, and language delay, with boys affected more than girls. These findings are in agreement with previously published studies [30,47,48] and may represent differences in brain organization and genetic or hormonal factors predisposing males to be more vulnerable to injury. We relate the lack of sex difference in autism in our cohort to the small number of patients with this diagnosis (*n* = 7). 

When comparing the outcome of Jews versus non-Jews, we found that most neurodevelopmental outcomes were poorer in non-Jews, despite a similar rate of follow-up visits in child development centers. In Israel, around 78% of the population are Jews, while Arabs comprise the largest minority group, around 21%, and Druze constitute a much smaller minority group, around 1.6% of the general population [49]. In general, rates of academic education and employment are higher in the Jewish population compared to the Arab population, along with higher incomes and socioeconomic status [50]. Healthcare ethnic disparities, mainly between Jews and Arabs, have been previously reported in studies utilizing large cohorts in Israel [51]. This difference may be explained by differences in socioeconomic status, health literacy, language barriers, and disparities in public healthcare and resources in distant rural areas. For example, one study utilizing the local Israeli national VLBW database found higher mortality in Arab VLBW preterm infants and less prenatal steroid use compared to Jewish VLBW preterms, likely reflecting inadequate timely access to perinatal services [52]. Genetics may also play a role, as well as consanguinity, which is more common in non-Jews [53]. Interestingly, there was a higher rate of DCD in Jews, which may be explained by higher awareness in this population of this diagnosis and better access to appropriate health services.

This study has collected data from patients born over a relatively long period of 10 years. Several changes in practice occurred over this time period, such as the growing use of non-invasive ventilation and minimally invasive surfactant administration worldwide, as well as the implementation of a national initiative for quality improvement that began in 2014 to reduce central line-associated bloodstream infection [54]. This initiative has shown suitable results with some reduction in infections at the national level (unpublished data); however, when analyzing rates of abnormal outcomes over the years in our cohort, these changes did not translate into consistent improvement in any of the neurodevelopmental outcomes.

Our study has several limitations. First, due to its retrospective nature, we could not account for many confounding variables such as maternal conditions, perinatal course, and postnatal exposures. We did not have available data on the family’s socioeconomic and parental education status. In addition, we were only able to collect data from the children who were followed in our child development centers, which introduces selection bias as VLBW preterm infants who were not CHS members could not have been evaluated in CHS’s child development centers. Furthermore, infants were referred to neurodevelopmental assessment at the discretion of the pediatrician so that subtle impairments may have been missed. Due to the retrospective cross-sectional nature of the study, we could only use the most recent available neurodevelopmental diagnosis and could not determine the average age at diagnosis. Due to the lack of current guidelines for developmental assessment, the children’s age at follow-up visits was not uniform. The strengths of this study lie in utilizing the large and reliable database of “Clalit” Health Services, which is the largest health service organization in Israel and one of the largest in the world, ensuring over 4.6 million people, and is by far the predominant health service organization in the Northern District of Israel. This allowed us to use a relatively large sample of patients who may likely be a representative sample reflecting the area’s population. Furthermore, all patients were at least 3 years old at the time of data collection, and 41% were over 6 years of age, thus increasing diagnostic accuracy compared to earlier evaluations. The importance of this study is the collection of our local neurodevelopmental outcomes for the first time on this scale.

In Israel, perinatal and postnatal information on preterm infants born <1500 g is collected in the national VLBW infant database, which is part of a collaboration of international neonatal networks, the iNEO (International Network for Evaluation of Outcomes). The database includes maternal and neonatal information to discharge, but no data after discharge are currently collected [55,56]. Israeli national guidelines determined by the Israel Ministry of Health for the post-discharge care of VLBW infants recommend at least six follow-up medical visits to a pediatrician or a neonatologist, physiotherapy or occupational therapy evaluation, and nutrition counseling, but there are no specific recommendations for a language and speech evaluation or for routine visits to the child development center. There is currently no collection, integration, or reporting of outcome data at a national level.

Our results should be interpreted within the framework of this study’s limitations. We have shown relatively high rates of CP and some group differences in neurodevelopmental outcomes in our population; however, causality cannot be detected in this cross-sectional retrospective design, and there was a lack of data on many confounding factors. Future well-designed prospective longitudinal studies should be performed to provide a more accurate picture of our local neurodevelopmental outcomes.

In light of our results, the authors feel it would be prudent to expand national data collection and post-discharge guidelines and to further assess the national healthcare resource allocation in Israel and the provision of post-discharge care to preterm infants.

## 5. Conclusions

Our study describes, for the first time, local neurodevelopmental outcomes of VLBW preterm infants after 3 years of age. We found a relatively high overall rate of CP and a significant difference in outcomes between Jews and non-Jews. This study emphasizes the need for an expanded and detailed local database of post-discharge outcomes and for assessment of national healthcare resource allocation and inequalities in the provision of care to preterm infants after discharge.

## Figures and Tables

**Table 1 children-10-01320-t001:** Demographic characteristics.

Variable		Number (Percent) N = 436
Sex	Male	220 (50.5)
	Female	216 (49.5)
Ethnicity	Jewish	248 (56.8)
	Non-Jewish	187 (42.8)
Mode of delivery	Vaginal	75 (17.5)
	Cesarean section	359 (82.5)
Multiple gestation		193 (44.3)
Birth weight, g (mean ± SD)		1132 ± 242
Weight percentile	SGA	89 (20.4)
	AGA	341 (78.2)
	LGA	6 (1.3)
Gestational age, weeks (mean ± SD)		29.5 ± 2.6
Gestational age	23–27 + 6 weeks	109 (25.0)
	28–31 + 6 weeks	230 (52.7)
	>32 weeks	97 (22.2)
Complications of prematurity		
	NEC	29 (6.7) *
	BPD	115 (26.5) *
	LOS	64 (14.7) *
	ROP	36 (8.3) **
	IVH grade 3	15 (3.4)
	IVH grade 4	9 (2.1)
	PHH	13 (2.9)
	WMI	25 (5.8)
	Cystic WMI	16 (3.6)

* N = 434, ** N = 432, SGA—small for gestational age, AGA—appropriate for gestational age, LGA—large for gestational age, NEC—necrotizing enterocolitis, BPD—bronchopulmonary dysplasia, LOS—late-onset sepsis, PDA—patent ductus arteriosus, ROP—retinopathy of prematurity, IVH—intraventricular hemorrhage, PHH—posthemorrhagic hydrocephalus, WMI—white matter injury.

**Table 2 children-10-01320-t002:** Neurodevelopmental outcomes.

Variable	Number (Percent) N = 436
No neurodevelopmental impairment	241/436 (55.1)
Cerebral palsy	39/436 (8.9)
Hemiplegic	8/436 (1.8)
Diplegic	13/436 (3)
Quadriplegic	14/436 (3.2)
Global developmental delay	53/433 (12.2)
Autism spectrum disorder	7/436 (1.6)
Attention deficit and hyperactivity disorder	34/436 (7.8)
Language delay	145/434 (33.4)
Developmental coordination disorder	16/434 (3.7)
Deafness	6/433 (1.4)
Blindness	2/432 (0.5)

**Table 3 children-10-01320-t003:** Neurodevelopmental outcome by gestational age.

Gestational Age (Weeks)	Normal Developmental Outcome	Cerebral Palsy	Global Developmental Delay	Autism Spectrum Disorder	Language Delay
23–27 + 6 weeks	43/109 (40.3)	22/109 (20.2)	24/109 (22)	3/109 (2.8)	49/109 (44.9)
28–31 + 6 weeks	136/230 (59.1)	13/230 (5.7)	21/230 (9.1)	4/230 (1.7)	65/230 (28.2)
>32 weeks	56/97 (57.7)	4/97 (4.1)	8/97 (8.2)	0/97 (0)	32/97 (33)

Number/total (percent).

**Table 4 children-10-01320-t004:** Neurodevelopmental outcome by abnormal sonographic findings in head ultrasound.

Neurodevelopmental Outcome	IVH				WMI				PHH		
	No IVH *n* = 364	Low-Grade IVH *n* = 48	High-Grade IVH *n* = 24	*p*-value	No WMI *n* = 411	Low-Grade WMI *n* = 9	Cystic WMI *n* = 16	*p*-value	No PHH *n* = 423	PHH *n* = 13	*p*-Value
CP *n* (%)	22 (6.0)	3 (6.3)	14 (58.3)	<0.0001	22 (5.4)	4 (44.4)	13 (81.3)	<0.0001	29 (6.9)	10 (76.9)	<0.0001
GDD *n* (%)	35 (9.7)	7 (14.6)	11 (45.8)	<0.0001	39 (9.6)	2 (22.2)	12 (75.0)	<0.0001	46 (11.0)	7 (53.8)	<0.0001
ADHD *n* (%)	27 (7.4)	4 (8.3)	3 (12.5)	0.876	31 (7.5)	2 (22.2)	1 (6.3)	0.255	32 (7.6)	2 (15.4)	0.269
Language delay *n* (%)	106 (29.2)	22 (46.8)	17 (70.8)	<0.0001	129 (31.4)	5 (51.6)	11 (68.8)	0.003	137 (32.5)	8 (61.5)	0.038
Autism *n* (%)	4 (1.1)	3 (6.3)	0 (0)	0.096	7 (1.7)	0 (0)	0 (0)	>0.99	7 (1.7)	0 (0)	>0.99
DCD *n* (%)	10 (2.8)	3 (6.3)	3 (12.5)	0.016	16 (3.9)	0 (0)	0 (0)	>0.99	16 (3.8)	0 (0)	>0.99

IVH—intraventricular hemorrhage (low grade—grade I/II, high grade—grade III-IV), WMI—white matter injury (low grade—periventricular echogenicities, high grade—cystic lesions), PHH—posthemorrhagic hydrocephalus, CP—cerebral palsy, GDD—global developmental delay, ADHD—attention deficit and hyperactivity disorder, DCD—developmental coordination disorder.

## Data Availability

The data presented in this study are available on request from the corresponding author. The data are not publicly available due to restrictions of privacy.

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
