# Peer review of "Neurodevelopmental Outcome of Very Low Birth Weight Infants in the Northern District of Israel: A Cross-Sectional Study"

_children, 2023, doi:10.3390/children10081320_

Round 1
Reviewer 1 Report
This manuscript outlines developmental outcomes of very low birthweight infants in an Israeli population between 2006-2016. While the subject matter is worthy of wider readership to contribute to the knowledge of outcomes of infants born with VLBW, there are some limitations which need to be addressed. The presentation of the results is hard to interpret given the aims of the study outlined. If the authors aim to describe differences between disease groups/within-group populations, then this needs to be described in the introduction. There is missing key information that would be pertinent to this population. There appears to be interchange between the terms “VLBW” and “preterm”. While there is crossover, less interchange between these terms would make the manuscript clearer. There is also a reliance on “statistical significance” without further attribution to odds ratios and confidence intervals. There is also specific information relating to follow up data and referencing missing from the manuscript.
Neurodevelopmntal outcome – what age? Lne 73
Objective tests and assessments quoted need to be referenced and justification of norm-reference to local population needed.
Cerebral palsy especially needs clearer definition on how the diagnosis was made and according to what consensus?
Table 1. while the population is VLBW, it would still be worthwhile quoting the mean birthweight for this population.
Table 1. percentage either needs to be whole number or one decimal point, not a mixture of both.
How was BPD determined/defined?
Were rates of cysytic PVL reported?
“Multiple logistic regression was used to identify the factors associated with compilcations of prematurity” – how is this achieved?
Major known contributors to poorer neurodevelopmental outcome are under-reported, e.g. cystic PVL, postnatal corticosteroids, neonatal surgery, lower socioeconomic status. Others are documented well.
Table 2: rates for CP doesn’t appear to be in line?
Line 118-119 does not seem to reflect findings in Table 3. It’s unclear which statistical test was used here. I assume logistic regression was used, but no odds ratio or confidence interval is quoted.
The presentation of rates of delay I nTable 3 according to apparently arbritrary categories of gestational age is interesting. It would be interesting to know why the data are presented this way.
The denominator changes between outcome variables as well. It is not apparently clear why.
There is a category of infants with gestation >35 weeks. What is the range of older gestational age? One would think that if an infant is classified as VLBW at >35 weeks, they are also affected by IUGR, yet this is not reported where I can locate it in the manuscript. The aim of the manuscript states that “we investigated the neurodevelopmental outcome of VLBW infants aged 3-13 years…”, yet there is no further aim outlined about gestational age categories defined by WHO as extremely preterm, very preterm, moderate-late preterm.
The aims state that the children were followed up between 3-13 years. What was the average age of follow-up? How often were they followed up? Which developmental outcome time point did they take?
If there are statistically significant difference in sex for rates of CP, which sex had a higher rate of CP? rates of sex, ADHD and language delay need to be demonstrated between sex so that the reader can ascertain if it is of clinical significance. This would be better quoted and described in terms of an odds ratio as well.
Table 4: what is low grade IVH? What is High grade IVH? Same with white matter injury grades. I do not know what PHH or PPH means. Please define CP, GDD, ADHD – the table needs to read as a standalone figure.
Figure 1 replicates information within the text. It does not enhance or add to the information. The legend of Figure 1 needs to be reversed. Presentation of differences in terms of odds ratio and confidence interval without the figure would be more useful information.
Discussion: Some of the information here needs to go in the introduction.
A 10-year recruitment period is wide. Changes in obstetric and neonatal care need to be considered in its effects on the results of this study.
Lines 178-189 – are these also Israeli studies? Comparison needs to be made with local populations or made clear if they are from different populations. It needs to be put into global or local context.
What age was CP diagnosed? Please comment on the age of diagnosis and stability of CP. Same with Autism, developmental delay, ADHD.
New data presented in lines 232 – would be good to know this in the results.
Reviewer 2 Report
Thank you for the opportunity of reviewing the manuscript “Neurodevelopmental outcome of very low birth weight infants in the Northern District of Israel: a cross-sectional study”. The authors conducted a necessary study following very low weight preterm infants in Israel’s northern district. Please find below some comments in order to improve the manuscript for possible publication.
1. Introduction section is really short, my suggestion is to expend it either in information about pre-term children or Israel’s Healthcare features.
Materials and methods.
2. Remove the 2.1, 2.2
3. Explain further the sociodemographic characteristics of the region.
4. Detail “major congenital abnormalities”
Results are well presented
Discussion:
5. Consider moving the first part of the discussion to the introduction section. Lines 161-177.
6. Discussion is well constructed, with interesting considerations about the results.
Conclusions: The conclusions in the abstract include a consideration about inequality. You may consider adding this also to the conclusions in the text.
Round 2
Reviewer 1 Report
Thank you to the authors for addressing the suggestions from the initial review. The manuscript is much clearer and has better flow. There are now some remaining minor issues that should be addressed as detailed:
Page 1 line 41
Rates of CP appear to be decreasing in some countries. Please review
Table 1 – C section rates seem high? Any comment on this?
Thank you for updating gestational age and including the mean +SD. I now notice that anyone between 27+1 to 27+6 are not reflected, and 32 weeks is included in the VPT category erroneously if it is according to WHO definitions.
Page 6 lines 214-224. Make sure the language used matches the odds ratio quoted. i.e. if the odds ratio is >1, the sentence needs to read “xx” was MORE LIKELY to be abnormal/normal than “xx” (OR xx, CI xx-xx); diagnosis of DCD in particular- odds ratio and sentence is currently confusing. Please also keep the hierarchy of reporting consistent in this section. Sometimes the p value appears first. Sometimes it is the odds ratio. Suggest prioritising OR, CI and then p-value. Suggest also avoiding the term “statistical significance” and replace with “strong evidence to suggest…”. There is also a bit of flipping between Jews and non-Jews values.
Line 299: “cystic WMI was most predictive of CP” – there are no predictive values quoted in your results. Suggest changing the word “predictive” to something less definitive, like “association” or similar.
Line 318: where the difference may be explained by a lack of “compliance”…compliance with what? I would be very careful to reconsider this word. There are multiple reasons why an ethnic group may be seemingly “non-compliant”. By labelling a lack of compliance, it assumes other negative generalisations about a group without truly understanding the actual barriers to health education and access.
The discussion needs to describe what the implications are from the limitations on the results of the study, or what future study designs are needed to provide a more accurate picture of the developmental outcomes.
Generally good, some minor errors.
